# The Effect of Surface Treatments on Zirconia Bond Strength and Durability

**DOI:** 10.3390/jfb14020089

**Published:** 2023-02-07

**Authors:** Dongni Shen, Huihua Wang, Ying Shi, Zhiwei Su, Matthias Hannig, Baiping Fu

**Affiliations:** 1Stomatology Hospital, School of Stomatology, Zhejiang University School of Medicine, Provincial Engineering Research Center for Oral Biomaterials and Devices, Zhejiang Provincial Clinical Research Center for Oral Diseases, Key Laboratory of Oral Biomedical Research of Zhejiang Province, Cancer Center of Zhejiang University, Hangzhou 310000, China; 2Department of Stomatology, Second Affiliated Hospital, Zhejiang University School of Medicine, Hangzhou 310009, China; 3Clinic of Operative Dentistry, Periodontology and Preventive Dentistry, Saarland University, D-66421 Homburg, Germany

**Keywords:** airborne-particle abrasion, glass-ceramic spray deposition, lithium disilicate glass ceramics, shear bond strength, zirconia

## Abstract

To evaluate the effects of airborne particle abrasion (APA) combined with MDP-containing resin cement, a glass-ceramic spray deposition (GCSD) method on the shear bond strengths (SBSs) and durability of 3 mol% yttrium oxide-stabilized zirconia ceramic (3Y-TZP) compared with lithium disilicate glass ceramics (LDGC). 3Y-TZP disks were randomly treated as follows: for Group APA+MDP, 3Y-TZP was abrased using 50 μm Al_2_O_3_ particles under 0.1 Mpa and bonded with MDP-containing resin cement; for Group GCSD, 3Y-TZP was treated with the GCSD method, etched by 5% HF for 90 s, silanized and bonded with resin cement without MDP. Group LDGC was bonded as the Group GCSD. X-ray diffraction (XRD), attenuated total reflection Fourier transform infrared spectroscopy (ATR-FTIR), X-ray photoelectron spectroscopy (XPS), scanning electron microscopy (SEM) and energy dispersive X-ray detector (EDX) were used to analyze the surface chemical and micro-morphological changes of the ceramics before bonding. The bonded ceramic specimens were randomly divided into subgroups, and the SBSs were determined before and after 10,000 thermocycling. The SBSs were analyzed with a one-way ANOVA analysis. Failure modes were determined with optical microscopy and SEM. The XRD, ATR-FTIR and XPS results identified the formation of lithium disilicate and zirconium silicate on 3Y-TZP after GCSD. The SEM micrographs revealed that 3Y-TZP surfaces were roughened by APA, while 3Y-TZP with GCSD and LDGC surfaces could be etched by HF to be porous. The APA treatment combined with MDP-containing resin cement produced the high immediate zirconia shear bond strengths (SBSs: 37.41 ± 13.51 Mpa) that was similar to the SBSs of the LDGC (34.87 ± 11.02 Mpa, *p* > 0.05), but, after thermocycling, the former dramatically decreased (24.00 ± 6.86 Mpa, maximum reduction by 35.85%) and the latter exhibited the highest SBSs (30.72 ± 7.97 Mpa, minimum reduction by 11.9%). The 3Y-TZP with GCSD treatment displayed the lower zirconia SBSs before thermocycling (27.03 ± 9.76 Mpa, *p* < 0.05), but it was similar to the 3Y-TZP treated with APA and MDP containing resin cement after thermocycling (21.84 ± 7.03 vs. 24.00 ± 6.86 Mpa, *p* > 0.05). The APA combined with MDP-containing resin cement could achieve the high immediate zirconia SBSs of those of the LDGC, but it decreased significantly after thermocycling. The GCSD technique could yield the immediate zirconia SBSs similar to those of LDGC before thermocycling, and long-term zirconia SBSs were similar to those of 3Y-TZP treated with APA followed by MDP-containing resin cement after thermocycling. Hence, the GCSD technique could enrich zirconia surface treatments and is an alternative to zirconia surface pretreatment for 3Y-TZP bond durability.

## 1. Introduction 

Zirconia ceramic has been widely used in prosthodontics [1] due to its favorable mechanical properties [2], biocompatibility [3], acceptable esthetics and chemical stability [4,5]. Silicon nitride and aluminum oxide are commonly applied ceramics materials in the medical implant; however, zirconia was reported to possess lower Tresca stress value under force loading and thus reduces the risk of postoperative failure [6]. Furthermore, the adhesion of resin cements to zirconia is still questionable due to its chemically inert surface and lack of micro-mechanical interlocking [7]. A resin-bonded fixed dental prosthesis made from zirconia exhibits more adhesive failure than that made from lithium disilicate [8,9]. Appropriate resin bonding protocols are crucial for the clinical success of zirconia restorations.

Airborne-particle abrasion (APA) followed by the application of 10-methacryloloxydecyl dihydrogen phosphate (MDP) has been reported to greatly improve the zirconia bond strength and durability of zirconia bonding to resin cement [10] because the APA could increase surface roughness and surface energy [11], and the latter facilitated the formation of P-O-Zr bonds by MDP [12,13]. However, APA has been reported to induce a phase transformation of the zirconia from a tetragonal crystal to a monoclinic crystal (t-m) structure at a pressure of 0.2–0.4 Mpa and cause a micro-crack formation under high pressure (0.4 Mpa) [13,14]. These effects may deteriorate the mechanical strength and compromise the long-term clinical performance [14]. Recently, APA with 50-μm aluminum oxide (Al_2_O_3_) particles at a reduced pressure of 0.1 MPa combined with MDP-containing resin cements has been recommended to provide durable bond strengths for zirconia [15,16]. Nevertheless, surface characteristics and micro-morphologies of zirconia under APA with such parameters were not investigated, and its bond strengths were not compared with those of glass ceramics in previous publications either [15,16]. 

Another generally accepted optional surface pretreatment for non-silica-based restorations is to increase the silica content combined with the help of silane and thus improve resin bonding [14]. Silanes are effective to promote the adhesion of silica-coated indirect restorative ceramics to resin composites with the formation of a siloxane linkage (-O-Si-O-)_n_ [14]. Pretreatment techniques of zirconia, including tribochemical silica coating, sol-gel methods, silicon nitride hydrolysis and vapor-phase deposition technique, have been reported to improve resin adhesion to zirconia [17,18,19,20], but the long-term bond performance of tribochemical silica coating is doubtful [18]. A condensed silanols layer with the sol-gel method that is hydrolyzed from tetraethyl orthosilicate [17] is prone to cracks under thermal treatment [21], and coatings via silicon nitride hydrolysis or vapor-phase deposition technique are time-consuming or demand complicated equipment; hence, it is impractical for clinical application [19,20]. Recently, a novel glass ceramic spray deposition (GCSD) method has been reported to improve the zirconia bond strength combined with 5% HF etching for 90–120 s when compared with an APA application at a pressure of 0.3 Mpa combined with an MDP-containing primer [22,23]. By spraying glass-ceramic powders on zirconia surfaces that are then sintered, a thin coating layer can be established [23] without affecting the physical properties of the zirconia [22]. MDP-containing resin cement has been recommended for bonding to zirconia because it achieved a higher bond strength of zirconia than an MDP-containing primer did [24]. Thus, the MDP-containing resin cement combined with APA application was adopted in this study when compared to GCSD method. Because CAD/CAM composites are highly polymerized [25], this might produce the comparatively low SBSs between resin cement and zirconia in the previous publication [23]. 

The objectives of this study were to evaluate the shear bond strengths (SBSs) and durability of resin cement to zirconia when zirconia surfaces were pretreated either with APA using a low pressure of 0.1 Mpa in combination with MDP-containing resin cement or with the GCSD method compared with those of a lithium disilicate glass ceramic (LDGC), as well as their surface characteristics and micro-morphologies.

The null hypotheses tested in this study were that there are no differences (1) in surface characteristics and micro-morphologies after zirconia surfaces were treated with APA or GCSD methods versus LDGC and (2) in their SBSs and bond durability.

## 2. Materials and Methods

### 2.1. Ceramic Disks Preparation

Ceramic disks (φ10 × 2 mm) were made of tetragonal zirconia polycrystals stabilized with 3 mol% yttrium (3Y-TZP) blocks (Superfect Zir, Aidite, China) or LDGC (Cameo, Aidite, China). Table 1 shows the materials used. The sintered 3Y-TZP disks were abraded using 50 μm Al_2_O_3_ (Hager &Werken GmbH & Co. KG, Germany) for 15 s under a pressure of 0.1 MPa at a distance of 10 mm (3Y-TZP with APA) [16]. The 3Y-TZP disks were treated with the GCSD method in accordance with the previous studies [22,23]. GCSD (Biomic LiSi connector, Aidite, China) was used to deposit a thin glass-ceramic coating layer on the surface of 3Y-TZP disks at a distance of 10 mm and sintered afterwards. The temperature was set at 450 °C at the initial stage, increased to 895 °C at steps of 80 °C /min, sustained for 1.5 min and slowly cooled finally. The sintered LDGC disks served as a positive control. The sintering steps were performed with a furnace (Austromat 624, DEKEMA Dental-Keramiköfen GmbH, Germany).

### 2.2. Surface Characterization of Ceramic Specimens

#### 2.2.1. XRD, ATR-FTIR and XPS Analysis

Because each crystal possesses a unique pattern of X-ray diffraction (XRD), XRD is widely used to characterize crystalline structure and for phase identification and transformation [26]. Because each functional group of molecules has a specific vibrating spectrum when detected by Fourier transform infrared spectroscopy (FTIR) [27], attenuated total reflection Fourier transform infrared spectroscopy (ATR-FTIR) is used to analyze the chemical compositions and functional groups of material surfaces [28]. Since X-ray photoelectron spectroscopy (XPS) can analyze binding energies of elements, XPS is used to determine qualitative information on elemental compositions and the valence state of the elements on the material surfaces [29]. The disks of 3Y-TZP, 3Y-TZP with APA, 3Y-TZP with GCSD and LDGC (n = 3) were examined with X-ray diffraction (XRD, X-pert Powder, PANalytical B.V., Holland) using Cu kα radiation (λ = 1.54 Å) with 2θ range of 5–90° at 0.026°/step and 19.89 s photon counting time per step and attenuated total reflection Fourier transform infrared spectroscopy (ATR-FTIR, Nicolet iS10, Thermo Scientific, USA) in a range of 4000 to 400 cm-1 in air at room temperature. In order to examine the influence of the GCSD coating layer on the 3Y-TZP, the 3Y-TZP disks and the 3Y-TZP disks (n = 3) treated with GCSD for 1–2 s at 25 cm distance were sintered and examined with X-ray photoelectron spectroscopy (XPS, AXIS Supra, Kratos, UK) equipped with a monochromatic Al Kα source (1486.7 eV of photons) under vacuum conditions (5 × 10^-10^ Torr). All the spectra were calibrated relative to the reference C 1s at 284.8 eV and subtracted with the Shirley method [30] with CasaXPS software 2.1.0.1 (Casa Software Ltd., UK).

#### 2.2.2. SEM and EDX Analysis

The surface morphology of 3Y-TZP, 3Y-TZP with APA, 3Y-TZP with GCSD and LDGC before and after HF etching (n = 3) was investigated with scanning electron microscopy (SEM, GeminiSEM 300, Carl Zeiss Microscopy GmbH, Germany). The elemental compositions were examined with energy dispersive X-ray analysis (EDX, XFlash 6-30, Bruker, USA). 

### 2.3. Bond Strength Testing

#### 2.3.1. Bond Procedure

Pre-cured composite resin cylinders (Filtek Z-250, 3M ESPE, St. Paul, MN, USA) were fabricated in transparent plastic tubes with an inner diameter of 3 mm and a height of 3 mm [12] using the incremental technique. Each layer of 1 mm thickness was light-cured with a light-curing unit with an intensity output of 1200 mW/cm^2^ (Elipar^TM^ S10, 3M ESPE, St. Paul, MN, USA). The composite resin cylinders were polished with 600-grit silicon carbide papers (Buehler, USA) under water coolant and cleansed ultrasonically in distilled water for 5 min before use. 

The 3Y-TZP disks (N = 64) were randomly divided into two groups (n = 32), and the LDGC disks (n = 32) served as positive control. The ceramic disks were processed and bonded according to the following surface treatments:

(1) Group APA+MDP: 3Y-TZP disks were pretreated with APA, as described in 2.1, ultrasonically cleaned with 99.5% ethanol for 3 min then totally dried with oil-free air spray before being bonded with resin cylinders using MDP–containing resin cement (Clearfil SA Luting cement, Kuraray Noritake Dental, Japan) by mixing equal amounts of paste A and paste B of the cement for 10 s. The bonded specimens were kept under a 5N load [31] for 3 min [32].

(2) Group GCSD: 3Y-TZP disks pretreated with GCSD were etched with 5% HF (IPS Ceramic Etching Gel, Ivoclar Vivadent IPS, Liechtenstein) for 90 s, thoroughly water-sprayed for 2 minutes, cleansed and then dried with oil-free air spray as mentioned above. They were then applied with silane agent (Monobond N, Ivoclar Vivadent IPS, Liechtenstein), left undisturbed for 60 s and strongly air-dried for 5 s. Finally, they were bonded with resin cylinders using resin cement without MDP (Variolink N, Ivoclar-Vivadent IPS, Liechtenstein) by mixing equal amounts of base and catalyst paste for 10 s. The subsequent bonding procedure was same as that of Group APA+MDP.

(3) Group LDGC: LDGC disks were etched by HF, applied with silane agent and bonded with Variolink N following the procedure of Group GCSD.

A schematic illustration of the materials and methods for bond strength testing is shown in Figure 1.

#### 2.3.2. Measurement of SBSs

Before SBSs testing, half of the specimens of each group were randomly divided into two ageing groups (n = 16), either stored in distilled water at 37 °C for 24 h or subjected to thermocycling at 5–55 °C with a dwell time of 30 s (Circulating Baths, MX20R, Polyscience, USA) for 10,000 cycles. 

The SBSs testing was performed with a universal testing machine (Instron5960, Instron Ltd, USA) with an error tolerance of ±0.5% at a crosshead speed of 0.5 mm/min [33]. The SBSs were measured in megapascals (MPa).

#### 2.3.3. Failure Mode Analysis

The failure modes of the fractured specimens were analyzed with an optical microscopy (Olympus BX53, Olympus Corp, Japan) and with SEM (GeminiSEM 300, Carl Zeiss Microscopy GmbH, Germany). The failure mode was classified as follows: for Type 1 (adhesive failure), cement was invisible on the fractured ceramic surface; for Type 2 (mixed failure), partial cement and partial ceramic residues were visible on the fractured surface; for Type 3 (cohesive failure), almost all of the fractured ceramic surfaces were covered with cement [29].

#### 2.3.4. Statistical Analysis

After data normality and homoscedasticity were determined with the Shapiro–Wilk test and Levene test, a statistical analysis of the SBSs was performed using one-way analysis of variance (ANOVA) and Tukey’s honestly significant difference (HSD) tests for multiple comparisons (SPSS Statistics 26.0, IBM SPSS, USA) (α = 0.05) [23]. 

## 3. Results

### 3.1. XRD, ATR-FTIR and XPS Analysis

XRD and ATR-FTIR results are illustrated in Figure 2. XRD pattern (Figure 2a) of 3Y-TZP presents tetragonal (t) and monoclinic (m) ZrO_2_ phases. The reflecting peaks of 3Y-TZP with APA (Figure 2b) exhibited similar XRD patterns as those of 3Y-TZP before APA, without an obvious increase of m-ZrO_2_ peaks. 3Y-TZP with GCSD exhibits t-ZrO_2_ and the Li_2_Si_2_O_5_ phase (Figure 2c). LDGC reveals the Li_2_Si_2_O_5_ phase (Figure 2d). The ATR-FTIR results for each surface are shown in Figure 2e. The ATR-FTIR spectra of the 3Y-TZP and 3Y-TZP with APA present strong broad bands at about 590 cm^−1^ and 602 cm^−1^, respectively, indicating the Zr-O stretches [34,35]. The ATR-FTIR spectra of the 3Y-TZP with GCSD and LDGC both present peaks at about 1100, 1000, 900, 780, 750, 440 cm^−1^, corresponding to the symmetric (ν_s_) and asymmetric (ν_as_) stretching (ν′_as_ (Si-O-Si), ν_s_ (O-Si-O), ν (Si-O-), ν_s_ (Si-O-Si), ν_as_ (Si-O-Si)) and the deformation of nonbridging oxygen Si-O-(Li^+^) bonds and Li-O symmetric stretching mode, respectively [36,37,38]. Spectral features between 500 and 560 cm^−1^ indicates the deformation vibration (δ (O-Si-O), δ (Si-O-Si) [36,37,38].

The binding energy changes of elements on the surface of 3Y-TZP disks before and after GCSD are shown in Figure 3. A wide spectrum of 3Y-TZP after GCSD (Figure 3b) shows atomic peaks of Si, O, Zr and C compared with 3Y-TZP of O, Zr and C (Figure 3a). Figure 3c displays the Zr 3d spectra of 3Y-TZP before and after GCSD. The binding energy peaks of Zr 3d_3/2_ and Zr 3d_5/2_ shifted to a higher value for 3Y-TZP with GCSD as well as an attenuation of peak intensity compared to the non-treated 3Y-TZP. Figure 3d–f displays the XPS spectra of 3Y-TZP with GCSD. They show typical peaks of O 1s at 531.9, 530.8 and 529.8 eV (Figure 3d); Si 2p at 102.3 and 101.8 eV (Figure 3e) and Zr 3d at 184.9, 184.3, 182.5 and 181.9 eV (Figure 3f), indicating the formation of Li_2_Si_2_O_5_ (O 1s at 531.9 eV, Si 2p at 102.3), ZrSiO_4_ (O 1s at 530.8 eV, Si 2p at 101.8, Zr 3d at 184.9 and 182.5 eV) and ZrO_2_ (O 1s at 529.8 eV, Zr 3d at 184.3 and 181.9 eV).

### 3.2. Surface Morphology and Elemental Composition

The EDX data and SEM micrographs of ceramic surfaces are shown in Figure 4. The 3Y-TZP disk surface presented Zr and O (Figure 4a). The representative SEM micrographs of the 3Y-TZP ceramic surface show a polycrystalline-grained structure without a glassy phase at high magnifications (Figure 4b,c). After APA treatment, the surface of 3Y-TZP was roughened and edge-shaped (Figure 4d,e). The 3Y-TZP surface with GCSD exhibited O, Si, Al and Nb (Figure 4f). The SEM micrographs of 3Y-TZP with GCSD showed smooth surfaces before HF etching (Figure 4g,h) and micro-gaps among approximately 3–10 µm of the crystals in the c-axis direction after etching (Figure 4i,j). The EDX data of LDGC showed O, Si and Al, and the Si amount on the GCSD and LDGC disk surfaces were similar (36.16 ± 3.46 wt% vs. 37.19 ± 1.36 wt%). The LDGC surfaces were flat before etching (Figure 4l,m) and revealed more numerous micro-gaps among approximately 1–2 µm of needle-like crystals in c-axis direction after HF etching (Figure 4n,o) compared with GCSD (Figure 4i,j). 

### 3.3. SBSs Testing

The mean SBSs values and their standard deviations (SD) are summarized in Table 2. Before thermocycling, there were no statistically significant differences in SBSs except for the comparison of group APA + MDP and group GCSD (*p* < 0.05). However, the group LDGC showed significantly higher SBSs than the groups APA + MDP and GCSD after thermocycling (*p* < 0.05). The group APA + MDP exhibited a higher amount of SBS reduction than the groups GCSD and LDGC (35.85% vs 19.20%, 11.90%).

### 3.4. Fracture Failure Mode Analysis

The distribution of fracture failure mode in Figure 5a showed that mixed failures and cohesive failures were the predominant failure mode in the specimens without thermocycling. After thermocycling, adhesive failures increased, cohesive failures were reduced and the mixed failure mode was predominant. Typical fracture failures of the 3Y-TZP with GCSD were displayed in Figure 5b because exposed zirconia could also be observed on the fracture surface accompanied by the remnants of GCSD coating layer on the surface (circles in Figure 5b).

## 4. Discussion

In this study, different elemental compositions were observed on 3Y-TZP, 3Y-TZP with GCSD and LDGC by EDX. The SEM findings revealed that 3Y-TZP with APA showed roughened and edge-shaped surfaces while HF etching produced porosities on 3Y-TZP with GCSD and LDGC surfaces (Figure 4). Thus, the first hypothesis (1) was rejected. The SBSs of the APA + MDP group were higher than those of GCSD group before thermocycling (*p* < 0.05) and comparable after 10,000 thermocycles (*p* > 0.05). The SBSs of GCSD group and LDGC group were statistically not different before thermocycling (*p* > 0.05) while the LDGC group showed higher SBSs than the GCSD group after 10,000 thermocycles (*p* < 0.05). Therefore, the second hypothesis (2) was also rejected.

APA combined with the chemical treatment of MDP-containing resin cement has been broadly accepted as an effective method for the cementation of zirconia restorations [10,12,39]. An Al_2_O_3_ particle abrasion is reported to increase the monoclinic volume proportion of zirconia [40]. The toughness of zirconia ceramic is related to the damage tolerance of zirconia along with the t-m phase transformation process [41]. Accompanied by 3–4% volume expansion, this transformation induces compressive stresses, close crack tips and, as a result, the prevention of further crack propagation [11]. However, with the progression of the monoclinic phase transformation from the zirconia surface to the bulk of the specimen, microcracks and tensile residual stresses may develop and reduce the flexural strength [41,42]. Okada et al. [11] suggested that the t-m phase transformation of zirconia crystals was induced at a pressure of 0.2–0.4 Mpa. Kwon et al. [43] further indicated the increase of m-phase along with the increase of Al_2_O_3_ particle size and pressure. Therefore, APA with a lower pressure is recommended to decrease the content of the monoclinic phase [44]. In this study, 50 μm Al_2_O_3_ particle abrasion at 0.1 Mpa led to a negligible monoclinic phase transformation (Figure 2) and yielded the highest SBSs before thermocycling when combined with MDP-containing resin cement (Table 2). The bond strength of 3Y-TZP treated with APA of 50 μm Al_2_O_3_ at 0.2 Mpa followed by the application MDP-containing primer and MDP-containing resin cement was reported to be similar to that of the lithium disilicate [45]. APA could not only roughen the zirconia surface (Figure 4), which increases micromechanical retention of the resin cement to the zirconia surface [46], but it could also increase zirconia surface energy that improves MDP binding to zirconia by P-O-Zr bonds [12,13]. The results in this study indicated that the APA using a reduced abrasion pressure in combination of MDP-containing resin cement produced the SBSs of zirconia as high as those of LDGC. However, the SBSs dramatically decreased after 10,000 thermocycles (Table 2), accompanied with an increase of the adhesive failure ratio (Figure 5). This might be the consequence of hydrolytic degradation of the P-O-Zr bond during water storage and thermocycling ageing [47]. This is also supported by the previous study that silane and SiO_2_ cluster is more stable in water than the MDP and tetragonal phase ZrO_2_ cluster [43], arising doubts on the long-term effectiveness of the zirconia bond.

The GCSD technique could produce a layer of glass-ceramic coatings with a thickness of approximately 11.78 μm on zirconia surfaces [23]. Previous studies have proved the improved bond strengths of GCSD on zirconia after etching with 5% HF for 90–120 s [22,23]. Herein, we demonstrated that 3Y-TZP with GCSD showed comparable SBSs with LDGC before thermocycling (Table 2), which is partly consistent with previously published data [22]. Paradoxically, Kang et al. [22] and Peng [23] reported that 3Y-TZP with GCSD showed higher SBSs than 3Y-TZP with APA using 50 μm Al_2_O_3_ at 0.3 Mpa combined with MDP-containing primer (Z-Prime Plus). However, the 3Y-TZP with APA + MDP treatment in this study produced higher SBSs before thermocycling compared to 3Y-TZP with GCSD. These contradictory findings might be explained by the different materials, MDP-containing primer (Z-Prime Plus), in previous studies while MDP-containing resin cement (Clearfil SA Luting cement) was adopted in this study. Yang et al. [24] proposed the application of MDP-containing resin cement in the clinic because it achieved a superior bond strength of zirconia compared to MDP-containing primer. Additionally, the SBS values of the APA+MDP group and of the GCSD group even after 10,000 thermocycles in this study are comparable to the immediate SBSs of APA and GCSD reported in previous publications [23]. This might be attributed to the highly-polymerized CAD/CAM composite cylinders that are not easily used to bond to zirconia [25]. 

The bond strengths and durability of 3Y-TZP with GCSD are determined by the two interfaces between 3Y-TZP and the coating layer, as well as between the coating layer and the resin cement. While the XRD and ATR-FTIR results indicated the successful establishment of the glass ceramic coating on 3Y-TZP (Figure 2), the XPS spectra (Figure 3) reveal the formation of zirconium silicate (ZrSiO_4_) between the coating layer and the zirconia surface [48,49,50]. The positive shifts of binding energy of Zr 3d (Figure 3c) are consistent with the previous publications [48,51]. This could be attributed to the t-m zirconia phase transformation or the formation of ZrSiO_4_ [50]. However, the XRD results of 3Y-TZP with GCSD (Figure 2) revealed no detectable m-zirconia phase. Thus, the positive shift of the Zr 3d binding energy should be attributed to the formation of ZrSiO_4_ [50] because the zirconium ion in zirconium silicate is more ionized than in zirconia [52]. The bind energy of 530.8 eV for O 1s, 101.8 eV for Si 2p, and 184.9 and 182.5 eV for Zr 3d fully coincides with previous results about ZrSiO_4_ [49,50]. The oxide dopants in the GCSD coating layer may play a crucial role in the formation of ZrSiO_4_ because the addition of metal ions, such as Li^+^, Nb^5+^ has been reported to lower the synthetic temperature of ZrSiO_4_ through the ZrO_2_–SiO_2_ system [53,54]. The chemical bond as well as the physical interlocking between zirconia and the GCSD coating layer are essential for the zirconia bond strengths [23]. After the establishment of the coating layer, EDX data (Figure 4) reveals that the Si amount of the coating is comparable with LDGC. This is also consistent with the previous publication [55]. The porosity of the etched layer (Figure 4), as well as the silane coupling through the formation of a siloxane (-O-Si-O-)_n_ network between the GCSD layer and the resin cement, might account for the similar bond strengths achieved by both 3Y-TZP with GCSD and the LDGC [56]. 

Regarding bond durability, the LDGC showed the highest SBSs after 10,000 thermocycles (Table 2). Water storage and thermocycling are widely used as artificial ageing methods to test the bond durability under laboratory conditions [57]. Since the chemical degradation of the bonding interface was caused by the moisture during thermocycling [58], the high bonding durability of LDGC can be explained by the surface porosity and, therefore, strong micromechanical interlocking (Figure 4). This is the reason that LDGC veneer could be successfully and widely used in dentistry. Considering that the SBSs of 3Y-TZP with GCSD were similar to those of 3Y-TZP with APA in combination with the MDP-containing resin cement after thermocycling ageing (Table 2), the GCSD technique might enrich the zirconia surface treatments and might be an alternative to APA pretreatment followed by the application of the MDP-containing primer or resin cement. Nevertheless, the degradation of the interface might furthermore be increased during clinical service. The limitation of the current study is the lack of cyclic fatigue that could outline the behavior of the adhesive cementation interface and the external gap progression even more [59,60]. Further clinical studies are also required to verify the long-term survival rate of zirconia restorations with different surface treatments.

## 5. Conclusions

3Y-TZP disks treated with APA, at a pressure of 0.1 Mpa, and MDP-containing resin cement produced the highest immediate zirconia SBSs, but the SBSs significantly decreased after thermocycling ageing. 3Y-TZP disks treated with GCSD produced SBSs comparable with LDGC disks before thermocycling ageing and with 3Y-TZP treated by APA and MDP-containing resin cement after thermocycling ageing. Thus, the GCSD technique can be used to enrich the zirconia surface treatment methods and is an alternative treatment for 3Y-TZP with durable bond strength.

Because the GCSD technique in this study could not achieve the same bond strength durability as LDGC, the GCSD should be further optimized. Cyclic fatigue that mimics chewing cycles should be carried out in a future study. A randomized clinical trial should be performed in the future to verify the effects of different surface treatments on zirconia bond strength and durability.

## Figures and Tables

**Figure 1 jfb-14-00089-f001:**
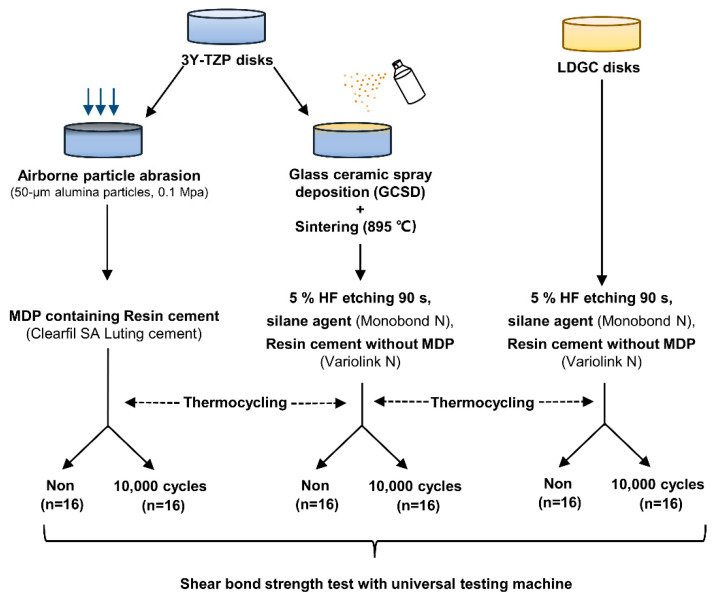
Schematic illustration of materials and methods for bond strength testing.

**Figure 2 jfb-14-00089-f002:**
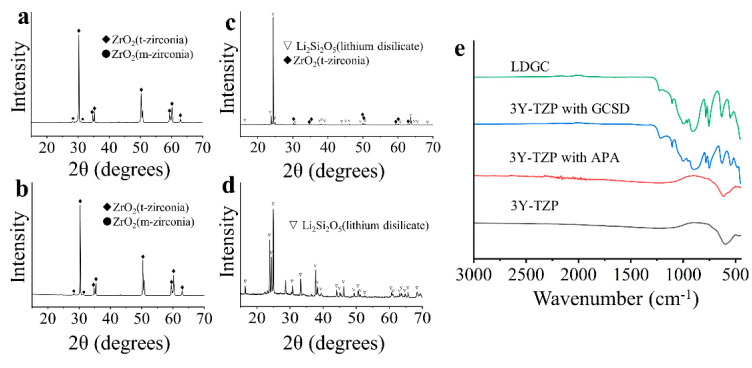
XRD (**a**–**d**), ATR-FTIR (**e**) spectra of 3Y-TZP, 3Y-TZP with APA, 3Y-TZP with GCSD and LDGC. XRD patterns of 3Y-TZP before (**a**) and after (**b**)APA show t-ZrO_2_ and m-ZrO_2_ phases. XRD patterns of 3Y-TZP with GCSD (**c**) displays t-ZrO_2_ and Li_2_Si_2_O_5_ phase, while LDGC (**d**) exhibits the Li_2_Si_2_O_5_ phase as well. ATR-FTIR spectra (**e**) illustrate the similar peaks of 3Y-TZP with GCSD and LDGC, hinting the successful establishment of glass-ceramic coating layer on 3Y-TZP by the GCSD method.

**Figure 3 jfb-14-00089-f003:**
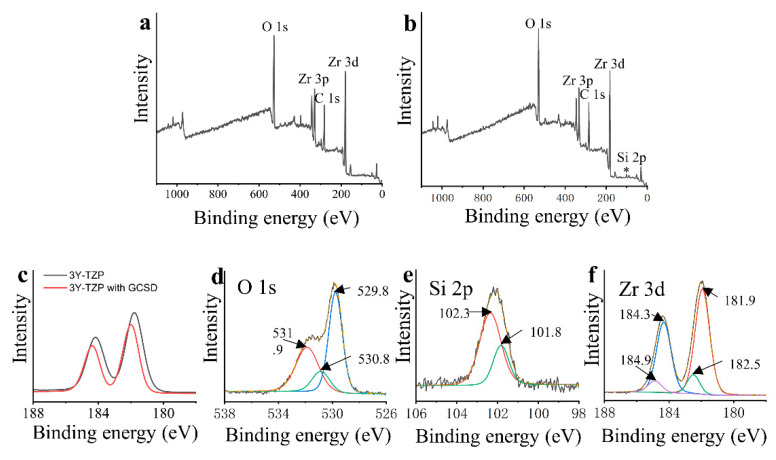
XPS spectra of 3Y-TZP before and after GCSD. The wide spectrum of 3Y-TZP (**a**) shows the main atomic compositions of Zr, O and C, while spectrum of 3Y-TZP after GCSD (**b**) exhibits additional Si peak (asterisk). (**c**) shows the peak shift and attenuation of Zr 3d peaks for 3Y-TZP with GCSD compared to the non-treated 3Y-TZP. (**d**–**f**) illustrate the XPS spectra of O 1s, Si 2p and Zr 3d of 3Y-TZP with GCSD, respectively.

**Figure 4 jfb-14-00089-f004:**
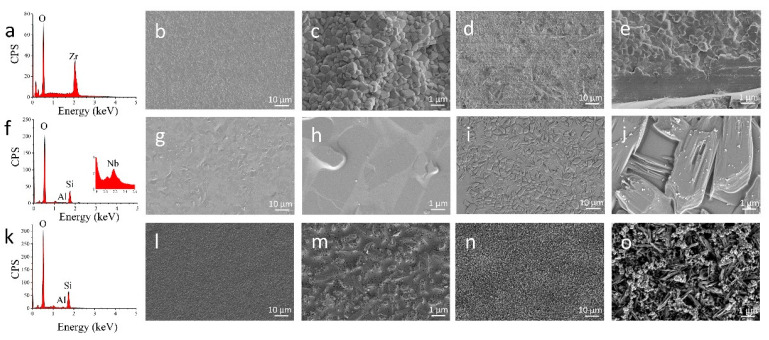
The EDX spectra (**a**,**f**,**k**) and SEM micrographs of 3Y-TZP (**b**,**c**), 3Y-TZP with APA (**d**,**e**), 3Y-TZP with GCSD (before etching (**g**,**h**), after etching (**i**,**j**)) and LDGC (before etching (**l**,**m**), after etching (**n**,**o**)). 3Y-TZP composed of Zr and O (**a**). 3Y-TZP with GCSD presented O, Si, Al and Nb (**f**). LDGC revealed the elements O, Si and Al (**k**). The 3Y-TZP disk showed typical polycrystalline-grained structure (**b**,**c**) and roughened surface after APA (**d**,**e**). The flat surface of 3Y-TZP with GCSD (**g**,**h**) and LDGC (**l**,**m**) revealed numerous porosities on 3Y-TZP with GCSD (**i**,**j**) and LDGC (**n**,**o**) after HF etching.

**Figure 5 jfb-14-00089-f005:**
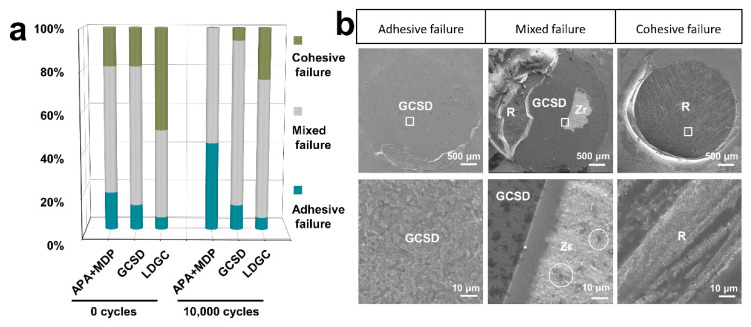
Fracture failure modes of resin cement bonding to 3Y-TZP with APA+MDP, 3Y-TZP with GCSD or LDGC. (**a**) Fracture failure mode distribution shows the failure proportions before and after thermocycling. (**b**) The typical fracture failures of 3Y-TZP treated with GCSD at low (×20, bar = 500 μm) and high (×1000, bar = 10 μm) magnifications. The resin cement residue (R) displays scratches since the pre-cured composite resin cylinders were polished before bonding. GCSD: glass–ceramic spray deposition; R: resin cement; Zr: zirconia.

**Table 1 jfb-14-00089-t001:** The materials used in this study.

Materials	Commercial Names	Compositions	Manufacturers (Country)	Lot Number
Zirconia ceramic	Superfect Zir	ZrO_2_ 94%–95 wt%, Y_2_O_3_ 4.5%–5.5 wt%	Aidite (China)	W200823NG-1
Lithium disilicate glass ceramic	Cameo	SiO_2_, Li_2_O	Aidite (China)	20210114-H-A2-1
Glass ceramic spray	Biomic LiSi connector	SiO_2_ 55%–60 wt%, Li_2_O 20%–25 wt%, Al_2_O_3_ 7%–10 wt%, K_2_O 9%–12 wt%, Nb_2_O_5_, etc.	Aidite (China)	20200710
Resin cement	Clearfil SA Luting cement	Paste A: Bis-GMA, TEGDEMA, MDP, hydrophobic aromatic dimethacrylate, silanated barium glass filler, silanated colloidal silica, dl-camphorquinone, benzoyl peroxide, initiator. Paste B: Bis-GMA, hydrophobic aromatic dimethacrylate, hydrophobic aliphatic dimethacrylate, silanated barium glass filler, silanated colloidal silica, surface treated sodium fluoride, pigments and accelerators	Kuraray Noritake (Japan)	4D0214
Hydrofluoric acid	IPS Ceramic Etching Gel	5% Hydrofluoric acid	Ivoclar Vivadent (Liechtenstein)	Z00DXK
Silane	Monobond N	Silane methacrylate, phosphoric methacrylate and sulfide methacrylate	Ivoclar Vivadent (Liechtenstein)	Z00DFM
Resin cement	Variolink N	Bis-GMA, urethane dimethacrylate, triethylene glycol dimethacrylate, ytterbium trifluoride, barium glass, Ba-Al-fluorosilicate glass, spheroid mixed oxide, stabilizers, pigments and initiators	Ivoclar Vivadent (Liechtenstein)	Z00647/Z00CV5

**Table 2 jfb-14-00089-t002:** The SBSs (Means ± SD, Mpa) of APA+MDP groups, GCSD groups and LDGC groups before and after thermocycling.

Groups	Thermocycling		
	0 Cycles	10,000 Cycles	Reduction
APA + MDP	37.41 ± 13.51 ^a^	24.00 ± 6.86 ^B^	35.85%
GCSD	27.03 ± 9.76 ^b^	21.84 ± 7.03 ^B^	19.20%
LDGC	34.87 ± 11.02 ^ab^	30.72 ± 7.97 ^A^	11.90%

Different letters in the same column indicate significant differences (*p* < 0.05). Same letters in the same column indicate insignificant differences (*p* > 0.05). Reduction refers to the percentage reduction rate of SBSs from 0 to 10,000 cycles.

## Data Availability

The datasets used and/or analyzed during the current study are available from the corresponding author on reasonable request.

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
