# Peer review of "The Effect of Surface Treatments on Zirconia Bond Strength and Durability"

_jfb, 2023, doi:10.3390/jfb14020089_

Round 1

Reviewer 1 Report

1.      In the abstract section, quantitative findings should be reported.

2.      Please add the abstract's "take-home" message, the current form was insufficient.

3.      Sort the keywords according to alphabetical order.

4.      Please do not use abbreviations in keywords.

5.      What is the novel of the present study? It works have been widely studied in the past. Nothing something really new in the present form. The lack of novel seems to make the present study like to replication/modified study. The authors need to detail their novelty in the introduction section. It is a major concern for rejecting this paper.

6.      It is essential to summarize previous studies' merits, novelties, and limitations in the introductory part to emphasize the gaps in the research that the latest research seeks to address.

7.      Line 70-76, please make a study’s objective to become narrative.

8.      The reasons of choosing zirconium oxide (zirconia) among others ceramics materials needs to be explain with comparison to others, such as silicon nitride and aluminium oxide as commonly ceramics materials applied in the medical implant. Also, additional relevant literature published by MDPI needs to be incorporated as follows: Minimizing Risk of Failure from Ceramic-on-Ceramic Total Hip Prosthesis by Selecting Ceramic Materials Based on Tresca Stress. Sustainability 2022, 14, 13413. https://doi.org/10.3390/su142013413

9.      To help the reader grasp the study's workflow more easily, the authors could include more visuals to the materials and methods section in the form of figures rather than sticking with the text that now predominates.

10.   It is required to include additional information on tools, such as the manufacturer, the country, and the specification.

11.   The revised manuscript after peer review must provide detailed information on the error and tolerance of the experimental equipment utilized in this study. Due to the disparate outcomes of other researchers' subsequent studies, it would make for a valuable discussion.

12.   Results must be compared to similar past research.

13.   Overall, discussion in the present article is extremely poor. The Authors must extend their discussion and make a comprehensive explanation. Just not simply mention the results with brief explanation.

14.   Please include the limitation of the present study, it is missing.

15.   Mention further research in the conclusion section.

16.   The reference needs to be enriched from the literature published five years back. MDPI reference is strongly recommended.

17.   The authors were encouraged to proofread their work due to grammatical problems and linguistic style.

18.   Provide graphical abstract for submission after revision.

Reviewer 2 Report

This paper is a very well-conducted study about the effects of airborne particle abrasion combined with MDP-containing resin cement, glass-ceramic spray deposition method on the shear bond strengths (SBSs) and durability of 3 mol% yttrium oxide-stabilized zirconia ceramic (3Y-TZP), compared with lithium disilicate glass ceramics.

The paper structure and the overall content are good. 

Nevertheless, I suggest some improvements be performed before this manuscript can be considered suitable for publication.

The authors could add a table with names, composition, type and Lot number of the materials used in this study.

Figure 4.: please add to the figure’s caption the meaning of the abbreviations: Cer, Zr and R. The reader could benefit this.

Use Discussion rather than Discussions

Line 300: After “thermocycling [45].” the authors could add a sentence that could outline a limitation of this study. Aging was obtained by thermocycling, but a simulation close to the clinical scenario can also be obtained through cyclic fatigue. The authors could therefore add a sentence like the following:

“Nevertheless, degradation of the interface could be furthermore increased during clinical service. One of the limitations of the current study is the lack of cyclic fatigue that could outline even more the behavior of the adhesive cementation interface and the external gap progression”.

To support this sentence, the authors could cite the following reference: https://doi.org/10.1111/jerd.12837 which compared zirconia and lithium disilicate interface through cyclic fatigue tests.

Please add other limitations of this study at the end of the discussion.

Please add further studies to be performed in the future on this topic.

Round 2

Reviewer 1 Report

Good job to the authors, but I have some other comments to response their revision as follows:

1.      In line 122-134, it is better to giving brief explanation about XRD, ATR-FTIR, and XPS first  for improve the understanding.

2.      Line 146-155 is preferred to make into point-by-point.

3.      Line 366-370, the authors explain the limitation due to cyclic fatigue, the additional relevant reference should be adopted as follows: Level of Activity Changes Increases the Fatigue Life of the Porous Magnesium Scaffold, as Observed in Dynamic Immersion Tests, over Time. Sustainability 2023, 15, 823. https://doi.org/10.3390/su15010823

Round 3

Reviewer 1 Report

I not have any further comments.

Author Response

We are grateful for your careful review and insightful comments that help us to improve the quality of our manuscript.